# Antioxidant and Anti-Inflammatory Effects of *Peganum harmala* Extracts: An In Vitro and In Vivo Study

**DOI:** 10.3390/molecules26196084

**Published:** 2021-10-08

**Authors:** Malik Waseem Abbas, Mazhar Hussain, Muhammad Qamar, Sajed Ali, Zahid Shafiq, Polrat Wilairatana, Mohammad S. Mubarak

**Affiliations:** 1Institute of Chemical Sciences, Bahauddin Zakariya University, Multan 60800, Pakistan; wasimchemist229@gmail.com (M.W.A.); Zahidshafiq@bzu.edu.pk (Z.S.); 2Institute of Food Science and Nutrition, Bahauddin Zakariya University, Multan 60800, Pakistan; Muhammad.qamar44@gmail.com; 3Department of Biotechnology, University of Management and Technology, Sialkot 51041, Pakistan; sajed.ali@skt.umt.edu.pk; 4Department of Clinical Tropical Medicine, Faculty of Tropical Medicine, Mahidol University, Bangkok 10400, Thailand; 5Department of Chemistry, The University of Jordan, Amman 11942, Jordan

**Keywords:** *Peganum harmala*, anti-inflammatory activity, antioxidant, toxicity, LC-ESI-MS/MS, traditional medicine

## Abstract

*Peganum harmala* (*P. harmala*) belongs to the family *Zygophyllaceae*, and is utilized in the traditional medicinal systems of Pakistan, China, Morocco, Algeria, and Spain to treat several chronic health disorders. The aim of the present study was to identify the chemical constituents and to evaluate the antioxidant, anti-inflammatory, and toxicity effects of *P. harmala* extracts both in vitro and in vivo. Sequential crude extracts including 100% dichloromethane, 100% methanol, and 70% aqueous methanol were obtained and their antioxidant and anti-inflammatory effects evaluated both in vitro and in vivo. The anti-inflammatory effect of the extract was investigated using the carrageenan-induced paw edema method in mice, whereas the toxicity of the most active extract was evaluated using an acute and subacute toxicity rat model. In addition, we have used the bioassay-guided approach to obtain potent fractions, using solvent–solvent partitioning and reversed phase high performance liquid chromatography from active crude extracts; identification and quantification of compounds from the active fractions was achieved using electrospray ionization mass spectrometry and high performance liquid chromatography techniques. Results revealed that the 100% methanol extract of *P. harmala* exhibits significant in vitro antioxidant activity in DPPH assay with an IC_50_ of 49 µg/mL as compared to the standard quercetin with an IC_50_ of 25.4 µg/mL. The same extract exhibited 63.0% inhibition against serum albumin denaturation as compared to 97% inhibition by the standard diclofenac sodium in an in vitro anti-inflammatory assay, and in vivo anti-inflammatory against carrageenan-induced paw edema (75.14% inhibition) as compared to 86.1% inhibition caused by the standard indomethacin. Furthermore, this extract was not toxic during a 14 day trial of acute toxicity when given at a dose of 3 g/kg, indicating that the lethal dose (LD_50_) of *P. harmala* methanol extract was greater than 3 g/kg. *P. harmala* methanolic fraction 2 obtained using bioassay-guided fractionation showed the presence of quinic acid, peganine, harmol, harmaline, and harmine, confirmed by electrospray ionization mass spectrometry and quantified using external standards on high performance liquid chromatography. Taken all together, the current investigation further confirms the antioxidant, anti-inflammatory, and safety aspects of *P. harmala*, which justifies its use in folk medicine.

## 1. Introduction

Free radicals are highly reactive species that can cause DNA damage owing to their unstable nature, and are a documented cause of different ailments including inflammation, cancer, ageing, bone diseases, cataracts, and even neurodegenerative disorders [1]. On the other hand, inflammation is the immune system response to injury, infection, or destruction characterized by heat, pain, swelling, redness, and disturbed physiological functions [2]. In contrast, chronic inflammation is usually related to pain, and involves some other events such as membrane disruptions, denaturation of protein, and an increase in vascular permeability [3]. Nonsteroidal anti-inflammatory drugs (NSAIDS) and corticosteroids are the most commonly used to reduce inflammation and relieve pain induced by inflammatory conditions [4]. However, extended exposure to these drugs may cause severe gastric lesions, digestive system upset, and liver and kidney diseases [5].

The genus *Peganum* is amongst the most significant medicinal plants, having two varieties of six species. *Peganum harmala* L., usually known as “Harmal”, is a glabrous perennial plant, which belongs to the family *Zygophyllaceae* [6,7]. *P. harmala* is a perennial herb growing in arid and semiarid regions around the globe including Mexico, southern regions of America, Africa, Pakistan, India, and various other countries [8]. *P. harmala* fruit and seeds have been used by the local herbal practitioners in Pakistan, especially in Bajaur Agency (tribal area of Pakistan) and Lakki Marwat, for the treatment of various diseases. Similarly, fruits are employed to relieve heart pain whereas seeds mixed with honey are used to treat fever, colic pain, and as virmifuge [9,10]. Indigenous communities of Wana, a district in South Waziristan Agency, Pakistan, used a decoction of *P. harmala* seeds to reduce irritation in the larynx, to fight against jaundice, as an abortifacient, to increase the flow of milk, and as a stimulant. In addition, seeds of *P. harmala* are known for their antiperiodic, antiparasitic, antispasmodic, narcotic, and galactagogic effects. In addition, *P. harmala* is used to treat asthma, colic, jaundice, and is found efficient in reducing fever in chronic malaria [11].

Moreover, *P. harmala* is used in the traditional medicinal systems of Morocco [12], Algeria [13], China [14], and Spain [15]. The plant extracts exhibit numerous biological activities such as anti-inflammatory, antibacterial, antifungal, antiviral, antioxidant, analgesic, cardio-protective, antitumor, antidiabetic, histofunctional, cerebral protective, anti-proliferative, and anticancer, among others [16,17,18,19,20,21]. Recently, *P. Harmala* leaf methanol extract was reported to exhibit antioxidant activities comparable to the standard quercetin and rutoside [22]. Research findings indicate that the alkaloids extract of *P. harmala* causes a significant antinociceptive effect in both phases of the formalin test in mice [23]. Similarly, *P. harmala* ethanol extract was found useful in the management of rheumatoid arthritis complications by boosting the intracellular antioxidant defense mechanism [24]. In addition, research findings demonstrate that *P. harmala* oil extract and seed alkaloid extract are not toxic in acute and subacute studies [25,26], however, the ethanol seeds extract of *P. harmala* shows toxicity in *Caenorhabditis elegans* [27].

In light of the previous discussion, it can be concluded that the *P. harmala* herb as a whole has not been investigated for safety aspects despite its use in different parts of the world. Accordingly, the aim of the present study was to evaluate the in vitro antioxidant and anti-inflammatory properties of the crude liquid–liquid partitioning, and RP-HPLC fractions of *P. harmala* obtained with the aid of bioassay-guided techniques. In addition, the present work highlights the safety aspects of *P. harmala* (the whole herb) through acute and subacute toxicity studies in rat models. To the best of our knowledge, these properties have not been fully investigated. Moreover, these tests were conducted to investigate whether the anti-inflammatory action of this plant in traditional medicine is justified.

## 2. Materials and Methods

### 2.1. Plant Material

Plant material was collected from DG Khan, in the district of Punjab, Pakistan and submitted to the Department of Botany, Bahauddin Zakariya University, Multan for taxonomic identification and authentication (Figure 1). A voucher specimen (ID 1132/PH) was deposited at the herbarium located at the pharmacy department, Bahauddin Zakariya University, Multan, Pakistan. The plant material was cleaned, washed, air-dried in the shade at room temperature, and then placed in an oven for 72 h at 37 °C, for complete dryness. It was then crushed, powdered by means of an electric grinder, and subjected to extraction processes.

### 2.2. Preparation of Extracts

The dried powder (2 kg) was extracted with *n*-hexane to remove fatty substances, and then sequential extraction was performed with 100% dichloromethane, 100% methanol, and 70% aqueous methanol using a temperature-controlled orbital shaker (Figure 2). The extract was filtered, and the filtrate was concentrated and dried by means of rotary evaporation (Heidolph, Hei-Vap, Schwabach, Germany) under reduced pressure (800 millibar) at 40 °C to afford semisolid crude extracts. Dried extracts thus obtained were stored at −18 °C in an upright ultralow freezer (Sanyo, MDF-U32V, Osaka, Japan) for future experiments.

### 2.3. Solvents and Reagents

Chemicals reagents and HPLC columns used throughout this investigation, including antioxidant reference/standards such as ascorbic acid, quercetin, ferrous sulfate, phosphate buffer, and dimethyl sulfoxide (DMSO); analytical grade solvents such as *n*-hexane, dichloromethane (DCM), chloroform, methanol, and water; HPLC grade solvents such as water, methanol, and trifluoracetic acid (TFA); and analytical and preparative HPLC columns (Zorbax-SB-C-18, Agilent, Santa Clara, CA, USA) were purchased from Sigma-Aldrich, St. Louis, MO, USA, and used as received.

### 2.4. Animals

In this investigation, we used 28 to 35 day old Wister albino mice (25–30 g) and Wistar albino rats (200–300 g). These animals were procured from the University of Lahore, Pakistan. Animals were housed in cages under standard conditions of 12:12 h light/dark cycle and 25 ± 2 °C. Animals were given free access to water and a standard diet (ad libitum) for 14 h prior to experiments, and kept under standard conditions mentioned in the Animals By-Laws N° 425–2008. All in vivo trials were performed according to the ethical codes set by the Institute of Laboratory Animal Resources Commission on Life Sciences, National Research Council (NRC, 1996), Washington, DC, USA. In addition, the animal care committee at Bahauddin Zakariya University, Multan, Pakistan, under identification number ACC-06-17, approved experimental assays.

### 2.5. Determination of Total Phenolic Content

We determined the total phenolic contents of the extracts according to the colorimetric Folin–Ciocalteu assay, which was also adopted by Hossain and Shah [28], using gallic acid as standard. Briefly, 0.5 mL of test sample was mixed with 1.5 mL Folin–Ciocalteu reagent and 1.2 mL of 7.5% sodium carbonate (Na_2_CO_3_) aqueous solution in test tubes. After an incubation period of 30 min in the dark, the absorbance of each test sample was measured at 765 nm using a spectrophotometer (UV-Vis 3000, Dresden, Germany). The phenolic content of each extract was expressed as mg gallic acid (GA) equivalent (E) per gram of dry extract (mg GAE/g); measurements were conducted in triplicate using ethanol as a blank.

### 2.6. Determination of Total Flavonoid Content

The total flavonoid contents of the extracts were determined using the aluminum chloride (AlCl_3_) assay as described by Oriakhi et al. [29] with slight modifications. Briefly, 0.5 mL of test sample was added to 0.5 mL distilled water, 0.15 mL of sodium nitrite (NaNO_2_) solution, and 0.15 mL of AlCl_3_ solution (2%). After 30 min incubation of the mixture in the dark, absorbance was determined at 510 nm with the aid of a spectrophotometer (UV-Vis 3000, ORI, Germany). Results were expressed as mg quercetin equivalents per gram (mg QE/g) of dry extract; determinations were conducted in triplicate using ethanol as a blank.

### 2.7. Determination of Antioxidant Activity

#### 2.7.1. DPPH Free Radical Scavenging Assay

The free radical scavenging potential of *P. harmala* crude extracts was determined according to the procedure outlined by Alara et al. [30] with slight modifications. Rather than using Soxhlet extraction and drying at 60 °C we used an orbital shaker for extraction and extracts were dried in an oven at 40 °C. In brief, 1 mL of test sample was mixed with 3 mL of 0.004% methanolic DPPH solution, and allowed to stand in the dark for 30 min. Then, absorbance of the reaction mixture was determined at 517 nm using a spectrophotometer (UV-Vis 3000, ORI, Germany). Quercetin and ascorbic acid were employed as standards, and methanol as the negative control. The percentage inhibition was evaluated using the following equation, and results were given as IC_50_:% inhibition = 100 × (A_C_ − A_S_)/A_C_,
where A_C_ = absorption of the control sample and A_S_ = absorption of the test sample.

#### 2.7.2. Ferric Reducing Antioxidant Power (FRAP)

The ferric reducing antioxidant power of *Peganum harmala* crude extracts was evaluated using the method of Zahin et al. [31] with some modifications. According to this method, 100 µL of extract was added to 300 uL of FRAP working solution (containing 300 mmol/L acetate buffer (pH 3.6), 10 mmol/L 2,4,6-tripyridyl-s-triazine (TPTZ) in 40 mmol/L HCl, and 20 mmol/L FeCl_3_ in a ratio of 10:1:1). After a 15 min incubation, absorbance of the reaction mixture was measured spectrophotometrically at 593 nm (UV-Vis 3000, ORI, Germany); ferrous sulfate was used for standard calibration. Results are expressed as mmol/g, and compared with ascorbic acid and quercetin.

#### 2.7.3. Hydrogen Peroxide (H_2_O_2_) Scavenging Activity

The hydrogen peroxide (H_2_O_2_) scavenging ability of *P. harmala* crude extracts was assessed as per the method of Ruch et al. [32] with slight modifications. H_2_O_2_ solution (40 mM) was added to 50 mM phosphate buffer (7.4 pH). Experimental extracts were then mixed with 0.6 mL H_2_O_2_ and incubated for 15 min. Absorbance of each mixture was recorded spectrophotometrically (UV-Vis 3000, ORI, Germany) at 230 nm. Ascorbic acid and quercetin were used as the positive control, and a phosphate buffer as the negative control. Percent inhibition of H_2_O_2_ was calculated using the following formula:H_2_O_2_ scavenging activity (%) = 100 × (A_C_ − A_S_)/A_C_
where A_C_ = absorption of the control sample and A_S_ = absorption of the test sample.

### 2.8. In Vitro Anti-Inflammatory Activity

#### 2.8.1. Membrane Stabilization Assay (Heat Induced Hemolysis)

In vitro anti-inflammatory experiments were performed by collecting blood samples from the cubital veins of healthy human subjects from Karachi, Sindh, Pakistan. All subjects voluntarily gave the blood samples after signing the consent performa. Blood samples were collected according to the guidelines of the International Federation of Blood Donor Organizations (IFBDO) with standard operating procedures, and were approved by the Bioethical Committee, Bahauddin Zakariya University, Multan Reg. no. 06–18. This work was also conducted in accordance with the Declaration of Helsinki. Blood samples were washed with normal saline after centrifugation (3000 rpm, 5 min) and then reconstituted as 10% *v*/*v* suspension with isotonic buffer solution (10 mM sodium phosphate buffer, pH 7.4) [33,34]. Finally, 1 mL of experimental samples of different concentrations (100, 200, and 300 µg/mL) were added to 1 mL (10%) of red blood cell suspension to make a reaction mixture of 2 mL, which was incubated for 25 min at 50 °C and then cooled to room temperature. After another centrifugation (2500 rpm; 5 min), absorbance of the reaction mixture was measured at 560 nm using a spectrophotometer; diclofenac sodium was employed as a standard drug and phosphate buffer as the control [35]. Doses were selected following the recently published research [36]. The percent inhibition was calculated using the following equation:% inhibition of denaturation = 100 × (A_C_ − A_S_)/A_C_
where A_C_ = absorption of the control sample, and A_S_ = absorption of the test sample.

#### 2.8.2. Egg Albumin Denaturation Assay

We conducted the egg albumin denaturation assay according to the procedure outlined by Mizushima and Kobayashi [37] with slight modification. In this method, 2 mL of a test sample with a concentration of 100–400 µg/mL was mixed with 0.2 mL of egg albumin and 2.8 mL of phosphate buffer (pH = 6.5) saline. The reaction mixture was incubated for 20 min at 37 °C followed by heating at 70 °C for 5 min. After cooling to room temperature, absorbance was measured at 660 nm with a spectrophotometer. We used diclofenac sodium as the standard drug and phosphate buffer as the control, and calculated the percentage inhibition according to the following equation:% inhibition of denaturation = 100 × (A_C_ − A_S_)/A_C_
where A_C_ = absorption of the control sample, and A_S_ = absorption of the test sample.

#### 2.8.3. Bovine Serum Albumin Denaturation Assay

The anti-inflammatory activity of the extracts was evaluated by the effect on bovine serum albumin denaturation; the rest was conducted according to the method described by Sakat et al. [34] with slight modifications. According to this method, a reaction mixture of 0.5 mL was made by mixing 0.05 mL experimental extracts (100–400 µg/mL) with 0.45 mL of bovine serum albumin and then incubated for 25 min at 25 °C. Afterwards, 2.5 mL of phosphate buffer (pH = 6.3) was added to the reaction mixture tubes and incubated in a water bath at 70 °C for 15 min. After cooling the mixture to room temperature, absorbance was measured at 660 nm with the aid of a spectrophotometer; diclofenac sodium was employed as a standard drug and phosphate buffer as the control. The percentage inhibition of protein denaturation was calculated using the following equation:% inhibition of denaturation = 100 × (A_C_ − A_S_)/A_C_
where A_C_ = absorption of the control sample, and A_S_ = absorption of the test sample.

### 2.9. In Vivo Anti-Inflammatory Activity 

#### 2.9.1. Inhibition of Carrageenan-Induced Paw Edema in Wistar Rats

We conducted the carrageenan-intoxicated paw edema study in accordance with the ethical codes set by the Institute of Laboratory Animal Resources, Commission on Life Sciences, National Research Council (NRC, 1996), Washington, DC, USA. Moreover, the animal use protocol was approved by the Institutional Animal Care and Use Committee at Bahauddin Zakariya University, Multan, Pakistan, under the protocol number AEC-06-18 and title: “In vivo anti-inflammatory activity of southern Punjab medicinal plants”. We employed the carrageenan-intoxicated inflammation model to evaluate the activity of *P. harmala* sequential crude extracts against inflammation following the method of Morris [38] with slight changes. In this method, *Wistar rats* were randomly divided into eight groups of five animals each (n = 5). Group 1 (control): rats were fed with normal saline. Group 2 (positive control): rats received standard indomethacin (100 mg/kg, b.w). On the other hand, rats in groups 3 and 4 were fed with 100 mg/kg and 200 mg/kg of DCM extract, respectively, whereas rats in groups 5 and 6 were fed with 100 mg/kg and 200 mg/kg of 100% methanol extract, respectively. Finally, rats in groups 7 and 8 were fed with 100 mg/kg and 200 mg/kg of 70% methanol extracts, respectively. In all these groups, the initial value of normal paw volume was measured. Doses were selected as per previous studies reported on *P. harmala* [8,39]. After 30 min of intraperitoneal administration of experimental extracts, freshly prepared 0.1 mL carrageenan in 0.9% normal saline was injected into the plantar aponeurosis surface of the right hind paw of each animal. Then, paw linear circumference was measured after 0, 1, 2, and 3 h of carrageenan injection by means of a plethysmometer (UGO-BASILE 7140, Comerio, Italy). The increase in paw circumference was considered as a way to measure inflammation.

#### 2.9.2. Inhibition of Formaldehyde-Induced Hind Paw Edema in Albino Mice

We have used the formaldehyde-induced hind paw edema in albino mice to evaluate the pain alleviating action of different extracts from *P. harmala* following Brownlee’s guidelines with slight modification [40]. In this method, albino mice were randomly divided into eight groups with five mice each (n = 5), and each group was treated according to the previous study. After 30 min of treatment with experimental extracts, 100 µL (4%) formaldehyde was infused into the plantar aponeurosis surface of the right paw of each mouse. Changes in the linear paw circumference were measured after 0, 3, 6, 12, and 24 h of formaldehyde infusion.

### 2.10. Acute and Subacute Toxicity Assessment

#### 2.10.1. Acute Toxicological Study

The study was performed by following the Organization for Economic Cooperation and Development (OECD) guidelines 407 and 423 for acute oral toxicity tests [41,42]. In this study, rats were given free access to clean drinking water and a standard diet (ad libitum) for 24 h before and after the experiment. *P. harmala* methanol extract doses of 1500 and 3000 mg/kg (per oral) were prepared as stock solutions for 14 days. After 1 h of extract administration, behavioral changes of each animal were noted periodically after 4, 8, 12, and 24 h. In addition, the body weight of all animals was recorded every 7 days, and samples of blood were collected for analysis for hematological and biochemical parameters. Afterwards, animals were sacrificed and organs (heart, liver, and kidney) were isolated for histopathological studies.

#### 2.10.2. Chronic Toxicological Study

All rats involved in this study were starved for 2 h before administration of 400 and 800 mg/kg (per oral) doses of *P. harmala* methanol extract for 28 days. After 1 h of extract administration, behavioral changes of each animal were recorded after 4, 8, 12, and 24 h. Furthermore, body weights of all animals were measured every weekend, and samples of blood were collected for analysis of hematological and biochemical parameters. At the completion of the study, all animals were sacrificed, and organs (liver and kidney) were isolated for histopathology.

#### 2.10.3. Histopathological Examination

Fresh organ portions from the heart, liver, and kidney, collected from normal and treated animals, were cut and fixed in 10% formalin solution. Fixed samples were dehydrated with alcohol dilutions series (60–100%) and embedded in paraffin. The paraffin fixed blocks of organs were cut to 4 µm thickness. These sections were stained with hematoxylin and eosin (H&E), and examined under light microscope for histopathological changes and photomicrographs were taken.

### 2.11. Liquid–Liquid Partitioning of Active Crude Extract

The 100% methanol extract was further separated using solvent–solvent partitioning by dissolving the crude extract with water, then chloroform, and finally with ethyl acetate. This extract was mixed with 15 mL of distilled water in a beaker and vigorously shaken with an identical volume of chloroform (15 mL). The two layers were separated: aqueous (top) and chloroform (bottom), and placed in different containers. The aqueous layer was again partitioned with chloroform and the process was repeated three times. The combined chloroform extracts were combined and evaporated under reduced pressure with the aid of a rotary evaporator to afford a semisolid thick paste and stored at −18 °C for future use. Using the same procedure, the aqueous layer was extracted with ethyl acetate. Both layers were recovered and evaporated using rotary evaporator. Importantly, the chloroform layer was named as fraction A, ethyl acetate fraction B, and the water layer as fraction C, and all fractions were evaluated for in vitro antioxidant and anti-inflammatory activities.

### 2.12. Method Optimization for Fractionation Using RP-HPLC

Fraction B, which showed significant antioxidant and anti-inflammatory activities, was further subjected to reversed phase column chromatography by dissolving solidified fractions into methanol as described by Cock [43]. Samples were prepared as 10 mg/mL and filtered using 0.45 mm syringe filter. The sample injection limit and flow rate were adjusted to 70 µL and 0.5 mL/min, respectively, using the Agilent LC technology and an analytical column (4.6 × 150 mm, 5 μm, Agilent, Waldbronn, Germany). Maximum number of peaks was observed with acidified (0.1% formic acid) water (A) and acidified (TFA) acetonitrile (B) at 254 nm.

Various combinations of mobile phases such as methanol: water, acidified methanol (0.1% TFA): acidified water (0.1% TFA), acidified methanol (0.1% FA): acidified water (0.1% FA), acetonitrile: water, acidified acetonitrile (0.1% TFA): acidified water (0.1% TFA), acidified acetonitrile (0.1% FA): acidified water (0.1% FA), and different wavelengths such as 210 nm, 230 nm, 254 nm, 280 nm, 300 nm, and 330 nm were used. A mixture of water (A) and acetonitrile (B), both containing 0.1% formic acid, was selected as the mobile phase for Fraction B of the *P. harmala* methanol extract. Then 10 µL sample was injected into the HPLC system and the linear eluting gradient was as follows: 10% B in 0–5 min, 10–40% B in 5–12 min, 40–60% B in 12–20 min, 60–80% B in 20–25 min, and 100% B in 25–30 min. Maximum number of peaks was observed with acidified (0.1% FA) water (A) and acidified (FA) acetonitrile (B) at 280 nm.

### 2.13. RP-HPLC Fractionation (Reversed Phase Chromatography)

Reversed phase chromatography was performed through a semipreparative column (C-18, 25 × 250 mm, 5 μm particle size, Agilent, Waldbronn, Germany). Samples were prepared as 50 mg/mL and filtered using a 0.45 mm syringe filter. The sample injection limit and flow rate were adjusted to 1 mL and 10 mL/min; this led to 5 subfractions: PHMF1, PHMF2, PHMF3, PHMF4, and PHMF5 from fraction B of the *P. harmala* methanol extract.

### 2.14. LC-ESI-MS/MS Analysis

We performed mass spectral analysis on RP-HPLC subfractions that have exhibited significant antioxidant and anti-inflammatory activity using LC-ESI-MS/MS (LTQ XL, Thermo Electron Corporation, Walthan, MA, USA) for tentative identifications of bioactive components according to the protocols suggested by Steinmann and Ganzera [44]. An online software was used to obtain the structures of bioactive compounds identified in the present study and to compare them with previously published data (www.chemspider.com, accessed on 6 October 2021).

### 2.15. Quantification of Compounds Using HPLC

*P. harmala* methanol fraction 2 (PHMF2) was dissolved in 1 mL of methanol in order to quantify the identified compounds using standard calibration curves. The mixture was centrifuged at 14,000 rpm for 10 min to collect the supernatant, whereas the filtration of supernatant was achieved using a syringe filter. Finally, we injected a 100-µL sample into the HPLC system for analysis.

### 2.16. Statistical Analysis

All determinations were conducted in triplicate and data were subjected to one-way analysis of variance (ANOVA). Results are expressed as the mean ± standard deviation (SD), and values are given as geometric mean with 95% confidence intervals (CI). Statistical analysis was performed using Dunnett’s test for significance at 95% confidence intervals (CI) with the aid of GraphPad Prism-6 (GraphPad Software, San Diego, CA, USA, http://www.graphpad.com, accessed on 6 October 2021); differences were considered significant at (* *p* < 0.05, ** *p* < 0.01, *** *p* < 0.001, **** *p* < 0.0001).

## 3. Results and Discussion

### 3.1. Phytochemical Constituents and Antioxidant Activity 

Phenolic compounds are plant secondary metabolites known to possess radical scavenging properties owing to their redox potentials. Results from this investigation show that the methanol extract has the highest amount of total phenolic (371.4 mg GAE/g) and flavonoid contents (1.3 mg QE/g) followed by DCM and hydro-alcoholic extracts as shown in Table 1. In addition, our findings reveal that the 100% methanol extract of *P. harmala* exhibits the highest antioxidant potential in the DPPH (IC_50_ 49 ± 3.1 µg/mL), FRAP (39 ± 0.9 mmol/g), and H_2_O_2_ (66% inhibition) assays as compared to other crude extracts. These results show a direct link between the quantity of phenolic compounds and antioxidant activity. The antioxidant activity of phenolics is mainly due to their redox properties, which make them act as reducing agents, hydrogen donors, and singlet oxygen quenchers. They also have a metallic chelating potential [45].

Published research indicates that the methanol seeds extract of *P. harmala* exhibits a higher free radical scavenging ability in DPPH assay (92% inhibition) than *n*-hexane, benzene, DCM, and chloroform extracts due to its higher phenolic contents of 30.9 mg GAE/g [46]; these results agree with findings of the present study. Recently, *P. harmala* leaf methanolic extract was reported to exert antioxidant activities in DPPH (IC_50_ 21.5 µg/mL) and FRAP (IC_50_ 32.4 mM TEAC/g) assays parallel to the standard quercetin outlined IC_50_ of 21.5 µg/mL (DPPH) and 32.6 mM TEAC/g (FRAP)) was also identified and quantified with a reasonable amount of phenolic compounds [22].

In addition, results displayed in Table 1 indicate that Fraction B exhibits stronger antioxidant potential as compared to others, such as fractions A and C, in all in vitro antioxidant trials. Similarly, PHMF2 of Fraction B (methanol extract) showed significant antioxidant activity, and was comparable to the antioxidant activity of the standard ascorbic acid and quercetin. This is in agreement with results obtained by Abderrahim and colleagues [21] using the DPPH assay. These researchers concluded that the antioxidant activity of *P. harmala* extract is primarily associated with their appreciable amount of flavonoids and polyphenols, which were calculated as 220.94 ± 1.1 and 650.38 ± 30.3 mg GAE/g.

### 3.2. In Vitro Anti-Inflammatory Activity 

#### 3.2.1. Heat Induced Hemolysis (Membrane Stabilization)

Agents that are capable of stabilizing the human red blood cell membrane in response to hypotonicity-induced lysis are recognized as anti-inflammatory drugs [47]. In this investigation, the membrane stabilization potential of *P. harmala* crude methanol extracts, liquid–liquid partitioned fractions, and HPLC fractions at different concentrations (100, 200, 300, and 400 µg/mL) was evaluated. Results reveal that PHMF2 exhibits the greatest inhibition (62.7%, *p* < 0.01) followed by Fraction B (55.3%, *p* < 0.01), and crude methanol extract (48.2%, *p* < 0.05), whereas the standard diclofenac sodium displayed potent inhibition (*p* < 0.0001) of 89.3% against heat-induced hemolysis in a concentration-dependent manner at 400 µg/mL as compared to the control (phosphate buffer), as shown in Table 2. Moreover, drugs that stabilize the red blood cell membrane may also protect the lysosomal membrane due to similar composition [48]. The in vitro anti-inflammatory activity exhibited by the methanol extract is consistent with its phytochemical and antioxidant potential.

Published research indicates that in vitro antioxidant, in vivo anti-inflammatory, and analgesic activities establish a beneficial role of the plant. In these findings, a cream formulation made from the seed oil demonstrated a significant anti-inflammatory effect. A slight peripheral analgesic effect was also observed due to the presence of polyphenols, linoleic acid, and γ-tocopherols, and is primarily due to antioxidant properties [49]. Earlier studies showed that plant extracts with antioxidant activity also display notable membrane stabilization properties in a concentration-dependent manner [50]. Previously, Khlifi et al. [51] studied *Artemisia herba-alba*, *Ruta chalpensis* L., and *P. harmala*, and also showed a direct relationship between phenolic compounds, antioxidant activity, and anti-inflammatory potential.

#### 3.2.2. Inhibition of Protein Denaturation (Serum and Egg Albumin)

Protein denaturation is an apparent cause of inflammation, and reliable literature cited earlier validates the link between inflammatory/arthritic problems and the denaturation of tissue proteins [52,53]. Results displayed in Table 2 show that *P. harmala* crude extracts and subsequent fractions exhibit considerable inhibition against egg albumin denaturation, where PHMF2 was the most potent (68.1%, *p* < 0.001), followed by fraction B (59.3%, *p* < 0.01), and crude methanol extract (57.1%, *p* < 0.01) at 400 µg/mL in a dose-dependent manner when compared to the control. These results are in accord with those obtained by Chandra et al. [2], which showed a concentration-dependent inhibition against denaturation of egg albumin by experimental coffee extracts and standard diclofenac sodium. Other researchers showed that *Enicostemma axillare* methanol extract at a concentration range of 100−500 µg/mL significantly protects the heat-induced protein denaturation [54]. Our findings show that PHMF2 (*p* < 0.001) exhibits notable inhibition against denaturation of serum albumin followed by fraction B (*p* < 0.001), and crude methanol extract (*p* < 0.01) with percentage inhibition of 72.9%, 71.7%, and 63.0%, respectively, at 400 µg/mL as compared to the control. In this respect, diclofenac sodium (*p* < 0.0001) showed significant inhibition against egg albumin and serum albumin denaturation at 400 µg/mL (Table 2).

### 3.3. In Vivo Anti-Inflammatory Activity of Sequential Crude Extracts

#### Carrageenan-Induced Paw Edema

Carrageenan-induced paw edema is an important method to investigate the anti-inflammatory potential of experimental extracts. Carrageenan-intoxicated paw swelling is a biphasic response wherein the first phase is associated with the release of pro-inflammatory cytokines such as kinins, histamine, and 5-HT, whereas the release of prostaglandins is associated with the second phase [55]. In the present study, *P. harmala* methanol extract, when dispensed at the rate of 200 mg/kg, causes notable inhibition (*p* < 0.001) of 75.14% after 3 h, which is similar to the 86.3% inhibition (*p* < 0.0001) caused by the standard drug indomethacin at 100 mg/kg when compared to the control (normal saline = 0% inhibition) (Figure 3). In contrast, the hydromethanol and DCM extracts were not as active. In this respect, *P. harmala* methanol extract could inhibit the mediation and release of pro-inflammatory cytokines in both phases of inflammation. The anti-inflammatory activities of the methanol extracts are consistent with their phenolic, flavonoids contents, and antioxidant activity. Our findings agree with those obtained by Edziri and colleagues [56]. These researchers found that both *M*. *alysson* and *Peganum harmala* methanol extracts exhibit good inhibition against carrageenan-induced paw edema; these extracts displayed good antioxidant activity and have high total phenolic content. In another investigation, the ethyl acetate extract of *P. harmala* caused significant inhibition of 70.3% at 200 mg/kg against carrageenan-induced paw edema. Preliminary phytochemical assessment of the extract indicated the presence of alkaloids and flavonoids in the extract [57]. Recently, *P. harmala* seeds ethanol extract, when dispensed orally at the rate of 100 mg/kg, was reported to reduce complete Freund’s Adjuvant intoxicated paw edema by up to 63.09% as compared to the control, and a decrease in synovial/hepatic tissues lipid peroxidation and increase in cellular antioxidants was also observed, supporting the findings of the current investigation [24]. In the present study, *P. harmala* methanol extract, when dispensed at the rate of 200 mg/kg, causes notable inhibition (*p* < 0.001) of 76.32% after 24 h, which is similar to the 89.3% inhibition caused by the standard drug indomethacin (*p* < 0.0001) at 100 mg/kg when compared to the control (normal saline = 0% inhibition) (Figure 4). In contrast, the hydromethanol and DCM extracts did not cause any substantial inhibition against formaldehyde-induced pain.

### 3.4. Acute and Subacute Toxicity Assessment

Our findings indicate that the *P. harmala* methanol extract exhibits substantial antioxidant and anti-inflammatory activities when compared to other extracts. Consequently, this extract was subjected to toxicity assessment. Results presented in Table 3 reveal no toxicity, body weight changes, behavioral changes, and mortality among animals during a 14 day trial of acute toxicity evaluation when the extract was given at a dose of 3 g/kg. Furthermore, the organ weights of the treated animals were within the normal range. Therefore, it was inferred that the LD_50_ of *P. harmala* methanol extract was above 3 g/kg. These findings are in agreement with those obtained by Selim et al. [25]. These researchers demonstrated no toxicity and no deaths in hamsters after 14 days of treatment with *P. harmala* oil extract at doses of 80, 160, and 320 mg/kg. Furthermore, *P. harmala* methanol extract was subjected to a 28 day trial of subacute toxicity. Results shown in Table 3 reveal no notable changes in the body and organ weights of the treated animals with 500 and 1000 mg/kg as compared to the control group (receiving normal saline). The weights of the heart, kidney, lungs, spleen, and liver of the treated animals were similar to those of the animals in the control group. Along this line, published work indicates that no clinical toxicity signs, changes in body weight, mortality, any gross morphological abnormalities in various organs of the treated mice, and relative weights of the organs were observed when *P. harmala* seed extracts were given to animals at a dose of 18 mg/kg [26].

#### 3.4.1. Hematological and Serum Parameters

Results pertaining to hematological and serum parameters of acute and subacute studies are presented in Table 3. Results revealed that in a 14 day trial, *P. harmala* methanol extract treated animals show a slight increase in the hematological and serum parameters, but within normal range. Similarly, *P. harmala* methanol extract revealed a slight increase in the hematological parameters when it was administered at the dose of 2000 and 3000 mg/kg as compared to the control group. In addition, results showed no adverse changes in hematological parameters in a 28 day trial of a subacute toxicity assessment of *P. harmala* methanol extract. However, there was a slight increase in parameters including WBCs, neutrophils, RBCs, hemoglobin, and platelets as compared to the control group (which received normal saline) when *P. harmala* methanol extract was given at the rate of 500 and 1000 mg/kg. Published work by Guergour and coworkers showed that treatment of female mice with alkaloids seeds of *P. harmala* did not cause significant changes in levels of urea and creatinine [26].

The serum profile in the acute and subacute studies did not show significant differences between normal and treated rats, which indicates that *P. harmala* methanol extract has no adverse effects on the liver and kidneys, and on biochemical parameters including serum total protein, albumin, lactate dehydrogenase, aspartate transaminase, total bilirubin, creatinine, and uric acid when given at the doses of 2000 and 3000 mg/kg as compared to the control group. Hence, consecutive oral administration of *P. harmala* methanol extract for 28 days has no toxic impact on the liver and kidneys (Table 3). In addition, research findings revealed that *P. harmala* does not cause significant changes in the hematological profile of female mice when compared with the control [26]; this agrees with findings obtained from this investigation that *P. harmala* is not harmful to the blood system.

#### 3.4.2. Histopathological Analysis

Displayed in Figure 5 are the histopathological sections of the heart, liver, and kidneys of the acute and subacute toxicity assessment. The microscopic observations revealed no substantial histopathological changes in *P. harmala* methanol extract treated rats as compared to those in the normal group. However, some lymphocytes with shrinkage of cardiomyocytes were observed in the heart muscles, which may indicate the presence of inflammation in subacute toxicity experiments. In kidney microscopy, interstitial edema and vacuolar degenerations were seen in subacute toxicity experiments. This indicates that renal epithelial cells inflammation was reversed by *P. harmala* methanol extract. Similarly, the architecture of the liver in the treated animals was similar to normal animals. Published work demonstrated that *P. harmala* in a subacute toxicity trial did not cause destruction to kidney architecture. Moreover, liver histology showed ground glass appearance of hepatocytes in acute toxicity [26].

### 3.5. ESI-MS/MS Analysis

Since PHMF2 demonstrated significant antioxidant potential as compared to the standards ascorbic acid and quercetin, and in vitro anti-inflammatory capacity as compared to the control, this fraction was subjected to ESI-MS-MS analysis for identification of bioactive compounds. Results revealed the presence of quinic acid, harmaline, harmol, harmine, and pegamine in the fraction as shown in Table 4 and Figure 6. To the best of our knowledge, quinic acid was reported for the first time in the *Peganum harmala* methanol fraction, whereas harmine and harmaline have been reported earlier. 

### 3.6. Quantification of Bioactive Compounds Using External Standards on HPLC

RP-HPLC subfraction PHMF2 was subjected to chromatographic investigation to further confirm and quantify the compounds identified based on mass spectroscopy using external standards (Table 5). External standards of quinic acid, peganine, harmol, harmaline, and harmine were well separated with our methodology, and eluted at retention times of 9.1, 10.1, 14, 21.8, 22.8, and 23.3 min at 280 nm as shown in Figure 7. Likewise, all these compounds were well separated from PHMF2 at the same retention times and wavelength, and quantified as 6.34, 19.2, 1.3, 3.9, and 53.9 µg/mg, respectively (Figure 8). Therefore, identities of these compounds were confirmed by comparing their retention times and mass spectral data with those of commercially available authentic samples as evident from HPLC overlay chromatograms (Figure 9). In this respect, Herraiz and coworkers reported a detectable amount of harmol, harmaline, and harmine as 0.03, 56, and 43.2 mg/g, respectively, from the *P. harmala* seed methanol extract [60]. On the other hand, the *P. harmala* methanol root extract also showed the presence of considerable amounts of harmol (14.1 mg/g) and harmine (20.6 mg/g), whereas harmaline was not detected. Other research groups indicated that harmol, harmaline, and harmine were present in the amounts 874, 488, and, 380 µg/mL, respectively, in the *P. harmala* methanol seeds extract [8]. Moreover, peganine was also identified and quantified earlier in the amounts of 9.6, 5.4, 21.2, 19.5, and 10.9 mg/g, respectively, from dry seeds, green fruits, immature fruits, flowers, and leaves of *P. harmala* [61]. In light of this, more work is required to identify and quantify more biologically active compounds from this traditionally important medicinal plant.

## 4. Conclusions and Future Prospects

In summary, the findings from this investigation reveal that extracts of *P. harmala* exhibit significant antioxidant and anti-inflammatory activities in both in vitro and in vivo models, which justifies the use of this plant in the traditional medicine of different countries. In addition, LD_50_ of PHME above 3000 mg/kg reflects the practical safety of the traditional herb with no changes in behavior. Most importantly, no mortality was observed during the acute (14 day) and subacute (28 day) toxicity studies. In addition, the presence of quinic acid, harmol, harmine, harmaline, and peganine may be responsible for the bioactivity of the extracts. However, more detailed studies are required to elucidate the possible mechanisms of action and pathways responsible for the antioxidant and anti-inflammatory capacities of the extracts. In conclusion, these results suggest that *P. harmala* possesses anti-inflammatory and antioxidant activities, and can be a promising new source of natural chemotherapeutic agents with no adverse effects.

## Figures and Tables

**Figure 1 molecules-26-06084-f001:**
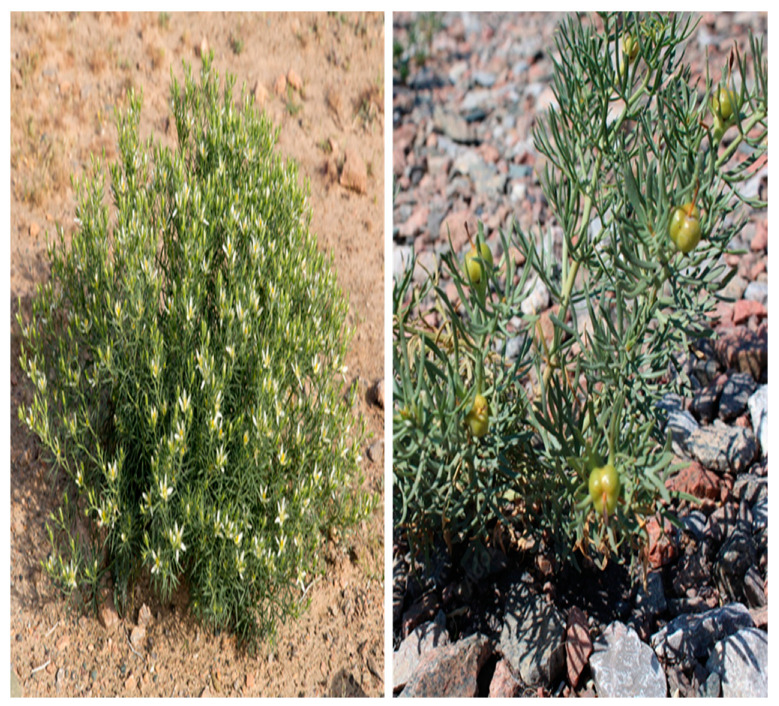
*Peganum harmala*.

**Figure 2 molecules-26-06084-f002:**
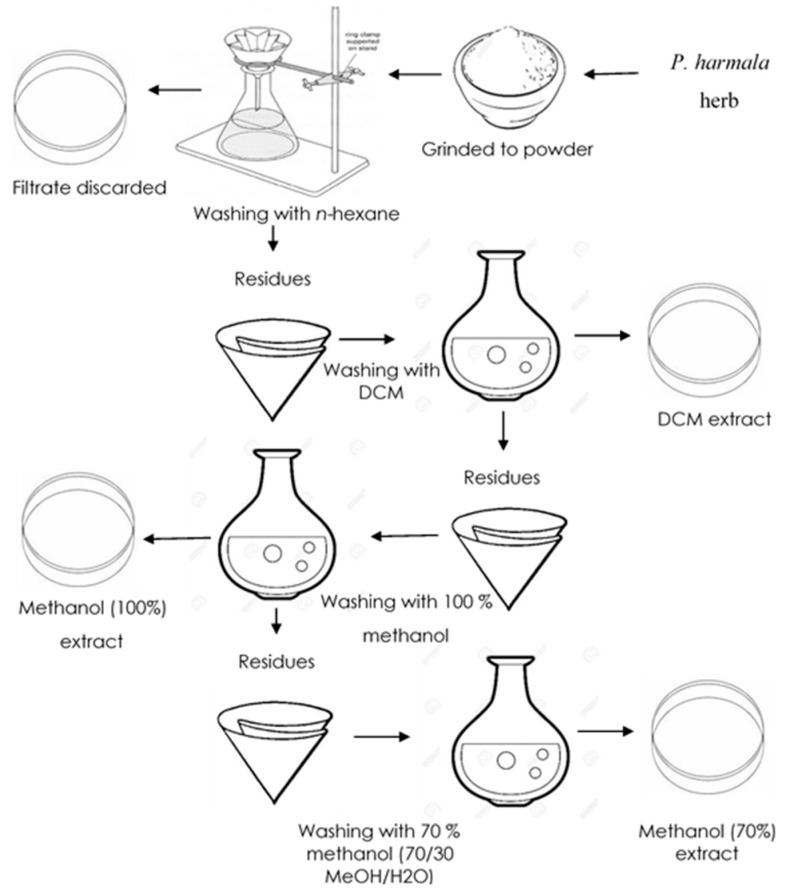
Sequential extraction of *P. harmala* powered using various solvents.

**Figure 3 molecules-26-06084-f003:**
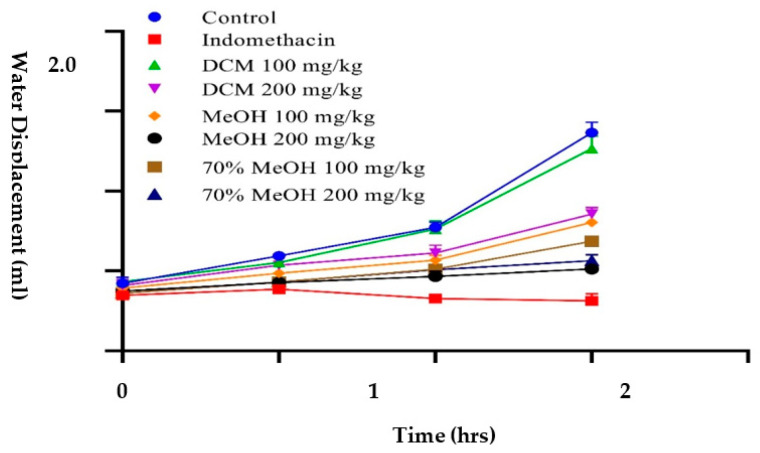
Inhibition against carrageenan-induced paw edema by *Peganum harmala* sequential crude extracts.

**Figure 4 molecules-26-06084-f004:**
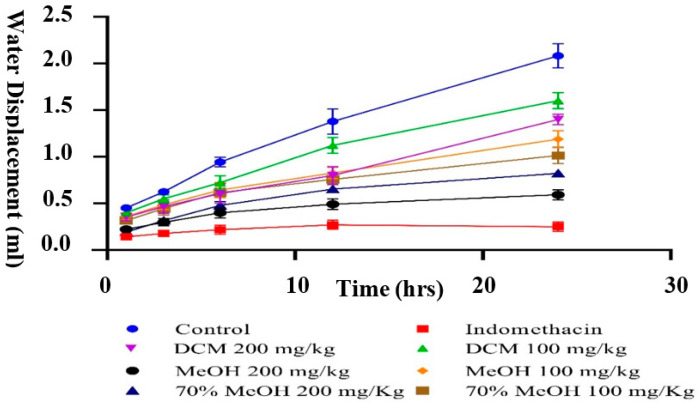
Inhibition against formaldehyde-induced paw edema by *Peganum harmala* sequential crude extracts. Values are mean ± SEM. *p* < 0.05 was considered significant (* *p* < 0.05, *** p* < 0.01, **** p* < 0.001, ***** p* < 0.0001).

**Figure 5 molecules-26-06084-f005:**
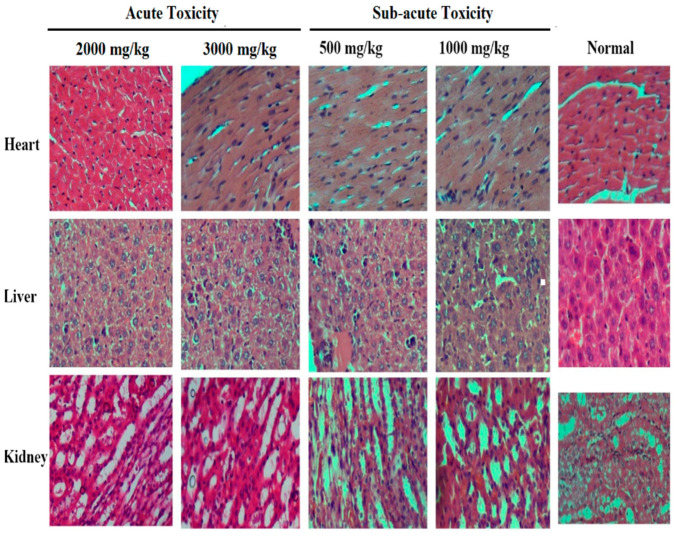
Histopathological investigation.

**Figure 6 molecules-26-06084-f006:**
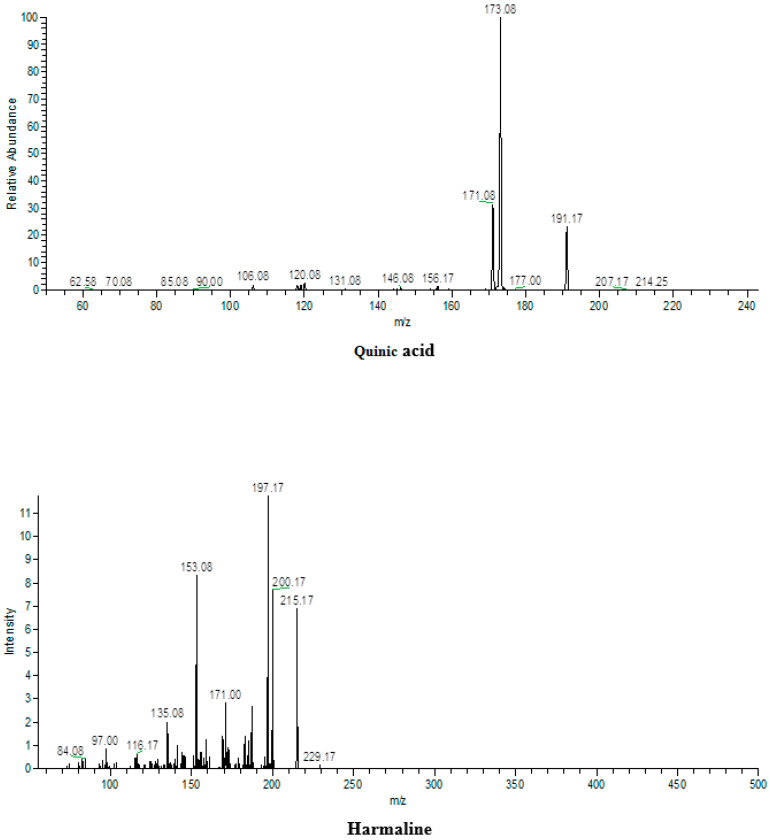
Mass spectra of different bioactive compounds found in **PHMF2** of *Peganum harmala*.

**Figure 7 molecules-26-06084-f007:**
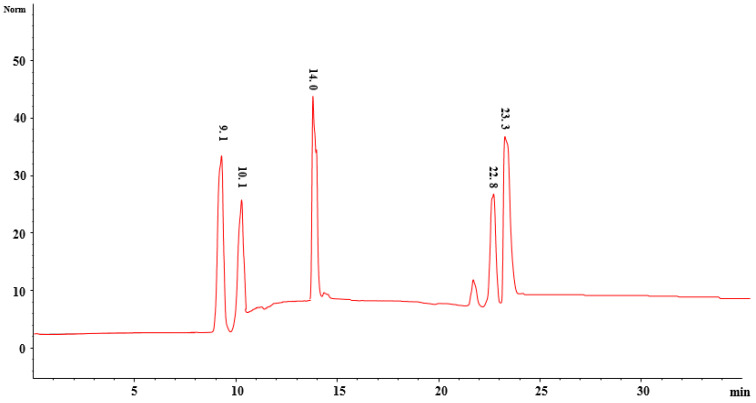
HPLC chromatogram of a mixture of external standards at 280 nm. (1) quinic acid, (2) peganine, (3) harmol, (4) harmaline, and (5) harmine.

**Figure 8 molecules-26-06084-f008:**
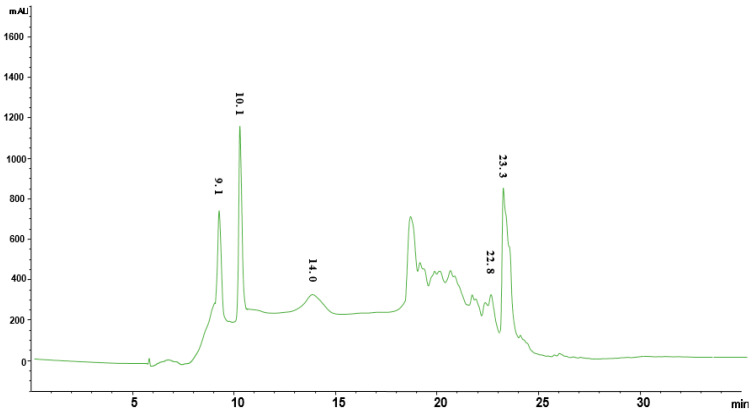
HPLC chromatogram of the PHMF2 subfraction at 280 nm. (1) quinic acid, (2) peganine, (3) harmol, (4) harmaline, and (5) harmine.

**Figure 9 molecules-26-06084-f009:**
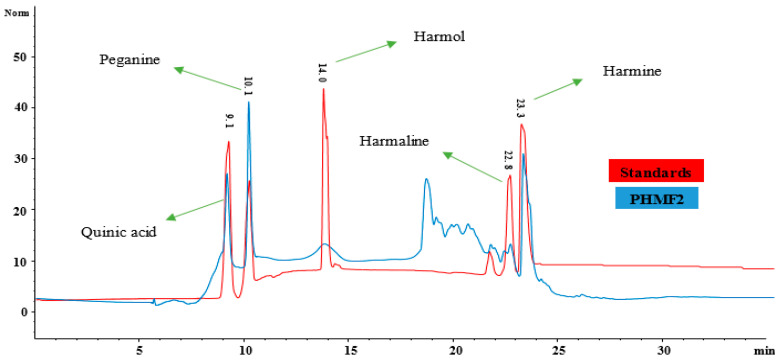
HPLC overlay chromatogram of the mixture of external standards and PHMF2 at 280 nm.

**Table 1 molecules-26-06084-t001:** Total phenolics and flavonoids content, and antioxidant potential of DCM, MeOH, and 70% MeOH extracts of *Peganum harmala* crude extracts and subsequent fractions.

Parameter	DCM Extracts	Methanol Extracts	70% Methanol Extracts	Fraction B (Methanolic Extract)	PHMF2 (Fraction B of Methanolic Extract)	Ascorbic Acid	Quercetin
**Total phenolic contents** **(mg GAE/g)**	106.2 ± 0.31	371.4 ± 0.2	142.3 ± 0.1	-	-	-	-
**Total flavonoid contents** **(mg QE/g)**	0.31 ± 0.5	1.3 ± 0.3	0.81 ± 0.02	-	-	-	-
**FRAP** **(mmol/g)**	9.2 ± 0.6	39 ± 0.9	19.2 ± 0.2	42.9± 0.1	45.3 ± 0.2	51 ± 0.02	62 ± 0.02
**DPPH** **(IC_50_ µg/mL)**	146 ± 2.0	49 ± 3.1	69 ± 1.4	44.6 ± 3.0	35.4 ± 1.1	29.1 ± 0.02	25.4 ± 0.01
**H_2_O_2_** **(%)**	25 ± 0.6	66 ± 0.9	43 ± 2.10	71 ± 2.0	75 ± 0.1	79 ± 0.02	84 ± 0.05

Values are means ± SD. DCM extracts = 100% dichloromethane extracts. MeOH = 100% methanol extracts. 70% MeOH = Methanol: water (70:50 *v*/*v*).

**Table 2 molecules-26-06084-t002:** In vitro anti-inflammatory activity of *Peganum harmala* crude extracts and subsequent fractions.

Treatment	Dose (µg/mL)	Membrane Stabilization		Egg Albumin Denaturation		Serum Albumin Denaturation	
		Absorbance	% Inhibition	Absorbance	% Inhibition	Absorbance	% Inhibition
**Control**	-	0.94 ± 0.1	-	0.91 ± 0.2	-	0.92 ± 0.1	-
**MeOH extract** (crude extract)	100	0.75 ± 0.2 ^ns^	20.2	0.69 ± 1.3 ^ns^	24.1	0.63 ± 0.3 *	31.5
	200	0.69 ± 0.4 ^ns^	26.5	0.65 ± 1.1 ^ns^	28.5	0.58 ± 0.2 **	36.9
	300	0.58 ± 0.5 *	38.2	0.56 ± 0.2 *	38.4	0.51 ± 0.2 **	44.5
	400	0.43 ± 0.1 **	48.2	0.39 ± 0.2 **	57.1	0.34 ± 0.1 **	63.0
**Fraction B** (liquid–liquid partitioned fraction)	100	0.66 ± 0.2 *	29.7	0.60 ± 0.2 *	34.0	0.58 ± 0.1 *	36.0
	200	0.65 ± 0.3 *	30.8	0.58 ± 0.2 *	36.2	0.54 ± 0.2 **	41.3
	300	0.55 ± 0.2 **	41.7	0.50 ± 0.2 **	45.0	0.48 ± 0.4 **	47.8
	400	0.42 ± 1.1 **	55.3	0.37 ± 0.2 **	59.3	0.26 ± 0.1 ***	71.7
**PHMF2** (RP-HPLC sub-fraction)	100	0.63 ± 0.1 *	32.9	0.59 ± 0.1 *	35.1	0.56 ± 0.2 *	39.1
	200	0.57 ± 0.3 *	39.3	0.54 ± 0.5 *	40.6	0.51 ± 0.2 *	44.5
	300	0.50 ± 0.2 **	46.8	0.49 ± 0.6 **	46.1	0.45 ± 0.1 **	51.0
	400	0.35 ± 0.2 **	62.7	0.29 ± 0.2 ***	68.1	0.25 ± 0.4 ***	72.9
**70% MeOH extract** (crude extract)	50	0.80 ± 0.4 ^ns^	14.8	0.76 ± 0.1 ^ns^	16.4	0.70 ± 0.3 ^ns^	23.9
	100	0.76 ± 0.5 ^ns^	19.1	0.72 ± 0.1 ^ns^	20.8	0.69 ± 0.2 *	25.0
	200	0.70 ± 0.4 *	25.5	0.65 ± 1.2 *	28.5	0.63 ± 0.2 *	31.5
	300	0.59 ± 0.1 *	37.3	0.55 ± 1.1 *	39.5	0.52 ± 0.3 **	43.4
**DCM** (crude extract)	100	0.94 ± 0.1	NA	0.94 ± 0.1	NA	0.93 ± 0.3	NA
	200	0.93 ± 0.1	NA	0.94 ± 0.2	NA	0.94 ± 0.2	NA
	300	0.93 ± 0.2	NA	0.93 ± 0.2	NA	0.94 ± 0.2	NA
	400	0.90 ± 0.2	NA	0.94 ± 0.1	NA	0.93 ± 0.3	NA
**Diclofenac sodium** (Standard drug)	100	0.19 ± 0.2 ***	79.7	0.16 ± 0.1 ***	82.4	0.05 ± 0.5 ****	94.5
	200	0.14 ± 0.2 ****	85.1	0.11 ± 0.1 ***	87.9	0.03 ± 0.1 ****	96.5
	300	0.11 ± 0.1 ****	88.2	0.09 ± 0.1 ****	90.4	0.02 ± 0.1 ****	97.0
	400	0.10 ± 0.2 ****	89.3	0.09 ± 0.5 ****	91.5	0.02 ± 0.5 ****	97.8

Values are means ± S.D. PHMF2; *Peganum harmala* methanolic fraction. 70% MeOH extract; methanol: water (70:50 *v*/*v*). ns; non significant (* *p* < 0.05, ** *p* < 0.01, *** *p* < 0.001, **** *p* < 0.0001).

**Table 3 molecules-26-06084-t003:** Acute and subacute toxicity assessment of *Peganum harmala* methanol extract.

Parameters	Control Group	Acute Toxicity (14-Day)		Subacute Toxicity (28-Day)	
	Normal Saline	2000 mg/kg PHME	3000 mg/kg PHME	500 mg/kg PHME	1000 mg/kg PHME
**Body weight (g)**	206.00 ± 4.50	208.00 ± 5.50	205.50 ± 6.33	209.83 ± 5.92	208.50 ± 7.17
**Organ Weight**					
Heart (g)	0.60 ± 0.15	0.65 ± 0.04	0.80 ± 0.06	0.74 ± 0.05	0.70 ± 0.10
Paired Lungs (g)	2.15 ± 1.15	2.53 ± 0.15	3.36 ± 0.47	3.14 ± 0.31	2.78 ± 0.81
Liver (g)	7.93 ± 1.40	8.10 ± 0.65	7.95 ± 1.33	9.66 ± 0.99	9.33 ± 1.37
Spleen (g)	0.45 ± 0.03	0.50 ± 0.03	0.47 ± 0.03	0.50 ± 0.03	0.67 ± 0.03
**Hematological Parameters**					
WBCs (10^5^/µL)	3.40 ± 0.02	2.03 ± 0.03	2.73 ± 0.02	3.07 ± 0.02	3.15 ± 0.02
Neutrophils (%)	40.20 ± 3.00	42.95 ± 4.50	43.55 ± 4.00	47.40 ± 4.25	48.40 ± 3.50
Lymphocytes (%)	45.45 ± 4.30	49.05 ± 5.50	49.45 ± 5.20	53.77 ± 5.35	56.57 ± 4.75
Eosinophils (%)	0.94 ± 0.10	0.96 ± 0.02	1.06 ± 0.05	1.09 ± 0.03	1.48 ± 0.07
RBCs (10^6^/µL)	9.05 ± 2.10	10.40 ± 0.50	11.35 ± 1.03	11.30 ± 0.77	12.27 ± 1.57
Hemoglobin (g/dL)	14.55 ± 2.45	16.05 ± 1.00	16.95 ± 1.27	17.12 ± 1.13	17.85 ± 1.86
Hematocrit (%)	45.60 ± 3.70	47.95 ± 3.50	48.75 ± 3.40	50.83 ± 3.45	56.43 ± 3.55
Mean corpuscular volume (MCV (f/L)	57.60 ± 7.50	62.05 ± 5.00	64.70 ± 5.43	66.88 ± 5.22	71.45 ± 6.47
Mean corpuscular hemoglobin (MCH (pg)	17.50 ± 1.65	18.80 ± 1.00	19.75 ± 1.03	19.72 ± 1.02	20.68 ± 1.34
MCHC (%)	29.45 ± 1.50	31.00 ± 2.00	32.25 ± 2.03	32.93 ± 2.02	33.90 ± 1.77
Platelets(10^5^/µL)	6.65 ± 1.10	6.95 ± 0.25	7.55 ± 0.53	7.58 ± 0.39	8.05 ± 0.82
**Serum Biological Parameters**					
Total Protein (g/dL)	4.05 ± 0.65	4.39 ± 0.25	5.10 ± 0.23	4.74 ± 0.24	6.51 ± 0.44
Albumin (g/dL)	1.88 ± 0.35	2.25 ± 0.20	2.45 ± 0.23	2.43 ± 0.22	2.69 ± 0.29
Albumin/Globulin ratio	2.30 ± 0.26	2.67 ± 0.03	2.95 ± 0.02	2.66 ± 0.02	2.75 ± 0.14
Lactate Dehydrogenase (U/L)	1732.00 ± 271.00	2088.50 ± 217.50	2266.50 ± 252.00	2281.00 ± 234.75	2379.00 ± 261.50
Asparate Transaminase (U/L)	111.30 ± 10.10	121.00 ± 14.50	141.10 ± 12.73	137.20 ± 13.62	136.47 ± 11.42
Alanine Transaminase (U/L)	29.85 ± 5.50	36.80 ± 4.50	39.35 ± 4.33	39.67 ± 4.42	43.33 ± 4.92
Alkaline Phosphatase (U/L)	284.00 ± 15.00	298.00 ± 19.00	312.50 ± 18.33	316.50 ± 18.67	320.17± 16.67
Total bilirubin (mg/dL)	0.25 ± 0.06	0.29 ± 0.01	0.36 ± 0.04	0.31 ± 0.03	0.29 ± 0.05
Creatinine (mg/dL)	1.43 ± 0.07	1.84 ± 0.45	2.46 ± 0.33	2.00 ± 0.39	1.77 ± 0.20
Uric Acid (mg/dL)	0.96 ± 0.06	1.15 ± 0.01	0.91 ± 0.04	1.04 ± 0.03	1.03± 0.05
Total Cholesterol (mg/dL)	53.70 ± 5.00	55.65 ± 3.50	60.20 ± 4.13	60.65 ± 3.82	66.52 ± 4.57
Triglycerides (mg/dL)	116.80 ± 9.40	121.95 ± 11.00	133.95 ± 8.77	133.00 ± 9.88	144.23 ± 9.08
Sodium (mmol/L)	108.00 ± 18.00	124.50 ± 11.00	125.00 ± 13.67	132.83 ± 12.33	139.17 ± 15.83
Chloride (mmol/L)	71.50 ± 20.00	91.50 ± 8.00	92.00 ± 12.33	97.33 ± 10.17	110.00 ± 16.17
Potassium (mmol/L)	3.04 ± 0.06	3.70 ± 0.40	3.07 ± 0.30	3.64 ± 0.35	3.44 ± 0.18

**Table 4 molecules-26-06084-t004:** LC-ESI-MS/MS identification of bioactive compounds from PHMF2 of *Peganum harmala*.

Fractions	Average Mass	ESI-MS/MS (Ions)	Compound	Chemical Formula	References
PHMF2	191	191, 173.1	Quinic acid	C_7_H_12_O_6_	[58]
	198	198.08, 181, 171.08	Harmol	C_12_H_10_N_2_O	
	213	213.17, 198.08	Harmine	C_13_H_12_N_2_O	
	214	215, 200.17, 197.17, 171	Harmaline	C_13_H_14_N_2_O	
	204	205, 187, 161	Pegamine	C_11_H_12_N_2_O_2_	[59]

**Table 5 molecules-26-06084-t005:** Quantification of quinic acid, peganine, harmol, harmaline, and harmine of PHMF2 obtained from *Peganum harmala* RP-HPLC subfraction.

Fractions	Compound Name	Wavelength	LOD	LOQ	*r* ^2^	R_t_ min	Concentration (µg/mg)
**PHMF2** **(Methanol extract)**	Quinic acid	280	1.1	3.2	0.9999	9.1	6.34
	Peganine		3.2	9.4	0.9989	10.1	19.2
	Harmol		0.2	0.5	0.9998	14.0	1.3
	Harmaline		0.4	0.9	0.9999	22.8	3.9
	Harmine		2.9	8.1	0.9997	23.3	53.9

Limit of quantification (LOQ), Limit of detection (LOD), Coefficient of regression (*r*^2^*)*, Retention time (Rt), *Peganum harmala* methanol fraction 2 (PHMF2).

## Data Availability

Available data are presented in the manuscript.

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
