# Peer review of "Antioxidant and Anti-Inflammatory Effects of Peganum harmala Extracts: An In Vitro and In Vivo Study"

_molecules, 2021, doi:10.3390/molecules26196084_

Round 1

Reviewer 1 Report

Dear Editor

The authors wrote a compelling and well-written research manuscript. Even though some methods are known, nonetheless the manuscript is written logically. 

  1. Abstract: Line 30. The authors were vague that the ethanol extracts results showed significant efficacy and yet no indication of the IC50 value compared to standard. Authors to add this information in the abstract.
  2. Line 59 remove have been  and rather use 'are'
  3. Line 66 remove has been  and rather use 'are'
  4. Line 75, caution, authors to refrain overuse of work also, you can still have a sound sentence without too much 'also'
  5. The methods used by authors are logical, flowing, quite pleasant for a reader.
  6. Figure 2. Diagram quiet blur, the word ' extraction is cut', authors to clean this
  7. Line 112-116, font size seemed small compare to the rest of the paper.
  8. Line 145, the authors indicated that ..with slight modification...what does that mean..as a scientist state the difference or the modification you are talking about.
  9. Line 289 and 292, Use 'we' once, a suggestion..an online was used.
  10. Authors are to be mindful of the words ‘them’, ‘they’ as they referred to a third person.
  11. Line 318 - used ' showed'
  12. Table 1: mg must be in one line, kindly clean up
  13. Table 2: Alignment is not straight
  14. Line 340, remove ' we have instead of at the end of the sentence add was evaluated.
  15. Line 347, remove' they'
  16. line 351, remove ‘have ‘
  17. line 370, join the sentences
  18. Figure 3 and 4 font sizes are over-exaggerated, reduce
  19. Figure 7 blur, change with one with better resolution
  20. Figure 8, same as above

Author Response

Reviewer 1

The authors wrote a compelling and well-written research manuscript. Even though some methods

are known, nonetheless the manuscript is written logically.

Sr #

Reviewer comments

Authors response

1

Abstract: Line 30. The authors were vague that the ethanol extracts results showed significant efficacy and yet no indication of the IC50 value compared to standard. Authors to add this information in the abstract

Added suggested information

2

Line 59, remove have been and rather use 'are'

Corrected as suggested

3

Line 75, caution, authors to refrain overuse of work also, you can still have a sound sentence without too much 'also'

Corrected as suggested

4

The methods used by authors are logical, flowing, quite pleasant for a reader

Thank you

5

Figure 2. Diagram quiet blur, the word ' extraction is cut', authors to clean this

Replaced with a better quality image

6

Line 112-116, font size seemed small compare to the rest of the paper

Corrected

7

Line 145, the authors indicated that ..with slight modification...what does that mean..as a scientist state the difference or the modification you are talking about

Information added

8

Line 289 and 292, Use 'we' once, a suggestion..an online was used.

Modified the sentence as suggested

9

Authors are to be mindful of the words ‘them’, ‘they’ as they referred to a third person.

Replaced the word ’they’ with ‘

Animals’

10

Line 318 - used ' showed'

Corrected as suggested

11

Table 1: mg must be in one line, kindly clean up

Corrected as suggested

12

Table 2: Alignment is not straight

Corrected as suggested

13

Line 340, remove ' we have instead of at the end of the sentence add was evaluated.

Corrected as suggested

14

Line 347, remove' they'

Corrected as suggested

15

line 351, remove ‘have

Corrected as suggested

16

line 370, join the sentences

Corrected as suggested

17

Figure 3 and 4 font sizes are over-exaggerated, reduce

Corrected as suggested

18

Figure 7 blur, change with one with better resolution

Replaced with better quality image

19

Figure 8, same as above

Replaced with better quality image

Reviewer 2 Report

The manuscript entitled Antioxidant and anti-inflammatory effects of Peganum harmala extracts: An in vitro and in vivo study contains new and important information with practical application to justify its publication. The paper is within the scope of the Journal. The presented research is well-planned, and the manuscript is generally well organized. Therefore, the Introduction provides some data on the stage knowledge of this issue. There were used an appropriate and modern methodology. The results are well presented and interpreted. The Iconography is appropriate. The discussions interpret the findings in view of the results obtained in this research. The authors have consulted numerous recent specialized references.

Therefore, the work could be of interest, but some points must be considered prior acceptance:

-Please provide all Latin words in italic type (ex. in vivo, in vitro).

-The scheme of the figure 2 must be corrected and completed.

- More details on the methodology are needed.

- More details on how were obtained the fractions A, B, C, as well as subfractions are needed.

- More technical details related to the HPLC analysis method - how the separate components were identified?

- Why only PHMF2 subfraction was HPLC analyzed?

- A comparative approach of the fractions and subfractions chemical profiles could help to explain the results obtained in antioxidant and anti-inflammatory tests. It could also help in statistical interpretation of the data obtained.

- The discussions could take in attention more other literature data related with this subject.

Author Response

Reviewer 2

Open Review

Comments and Suggestions for Authors

The manuscript entitled Antioxidant and anti-inflammatory effects of Peganum harmala extracts: An in vitro and in vivo study contains new and important information with practical application to justify its publication. The paper is within the scope of the Journal. The presented research is well-planned, and the manuscript is generally well organized. Therefore, the Introduction provides some data on the stage knowledge of this issue. There were used an appropriate and modern methodology. The results are well presented and interpreted. The Iconography is appropriate. The discussions interpret the findings in view of the results obtained in this research. The authors have consulted numerous recent specialized references.

Therefore, the work could be of interest, but some points must be considered prior acceptance:

-1. Please provide all Latin words in italic type (ex. in vivoin vitro).

Response: Corrected as suggested

-The scheme of the figure 2 must be corrected and completed.

Response: Corrected as suggested

- More details on the methodology are needed.

Response: Information added

- More details on how were obtained the fractions A, B, C, as well as subfractions are needed.

Response: Complete information about liquid-liquid partitioning added.

- More technical details related to the HPLC analysis method - how the separate components were identified?

Response: Complete information method optimization and separation added

- Why only PHMF2 subfraction was HPLC analyzed?

Response: Only PHMF2 sub-fraction was HPLC analyzed because it showed in vitro antioxidant and in vitro anti-inflammatory activity better than other sub-fractions and also in accordance with standards

- A comparative approach of the fractions and subfractions chemical profiles could help to explain the results obtained in antioxidant and anti-inflammatory tests. It could also help in statistical interpretation of the data obtained.

Response: Respective reviewer you are right, we also adopted a comparative approach and relaying on that we screened crude extract (s), then liquid-liquid partitioned fractions, and finally RP-HPLC sub-fractions

- The discussions could take in attention more other literature data related with this subject.

Response: Discussion has been revised and more data added

Reviewer 3 Report

The manuscript entitles “Antioxidant and anti-inflammatory effects of Peganum harmala extracts: An in vitro and in vivo study” investigated that the potential of Peganum harmala extract, their bio constitutes, and finally their bioactivities both in-vitro and in-vivo.  The subject frame of the work is well constructed. So, in this respect and this article should be contributed to present research. I recommended this work re-consideration for publication after the following major revisions.

  1. The manuscript written very poorly, sometime its difficult to fellow the story. I recommend to use of proper English grammar. Also, there are several typographical mistakes as well in whole manuscript. Therefore, the author’s thoroughly careful check the language and typo mistake to minimize the error.
  2. The abstract should be beginning with a sentence about the background of concept and the aims as well as novelty of study should be mentions. What exactly is the novelty of this study? The abstract is poorly written and should be improved. Abbreviations must be avoided in abstract. Parenthesis should be avoided in abstract - this is poor writing. Please improve.
  3. Introduction; Check and format the citations in the whole manuscript. Also, Appropriate references must be provided to explained the background, what is already done and why this study carried out. Other vise the novelty of this research is still poorly presented. This is important especially for the high IF journals. The scientific style should be used. What exactly is the aim of this work? Hypothesis statement is missing in the introduction section.
  4. Introduction; Introduction section should be elaborated with recent articles on antioxidant and anti-inflammatory activities. Also about extract preparation as well as the importance of medicinal plant is missing. For example; the recommended literature are as fellow; https://www.sciencedirect.com/science/article/pii/S0045206815300031;
    https://www.sciencedirect.com/science/article/pii/S0141813021011818;
    https://www.sciencedirect.com/science/article/pii/S0278691517301035;
    https://www.sciencedirect.com/science/article/pii/S2213343721002670;
    https://www.sciencedirect.com/science/article/pii/S1570023218305749;
  5. Material and methods; The whole M&M section is poorly written and must be substantially rewritten and improved. The methods are not properly referenced and are not possible to follow, reproduce and verify. This is serious drawback of this research work. You must be precise when writing protocols (M&M section); everyone should be able to repeat your experiments, after this paper is published, and gain the same results. Serious lack of relevant information or poorly written Materials and methods section are always an alarm form me, as a reviewer. For example, Materials are poorly described. Furity and product codes of all these materials should be provided. Materials are poorly described. Product codes should be provided. Appropriate references are missing. Who exactly is the Author of the methods applied? Are these methods valid/correct? Without sufficient information it is hard to reproduce the results, follow, and compare the idea and the obtained results with the literature. Appropriate references to the methods should be provided.
  6. The statistical findings have to be given in the text such as (p<0.05) or (p>0.05).
  7. Results and discussion; General remark to the discussion - In my opinion, the discussion provided by Authors is difficult to follow and verify due missing critical details in the methodology section. Due to poorly described material and poorly presented methods, I am not able to follow and properly review the discussion.
  8. All figures are of poor technical quality and not suitable for publication, especially in a high reputed journal. Font size and kind is too small and must be unified in all figures. Small writings are unreadable. All figures must be self-explanatory. Axis titles are poorly presented or absent. Units are missing. Are the data presented in figures significantly different? At least error bars should be shown.
  9. Details about the statistical testing are needed.
  10. I suggest first time write full name rather than abbreviation; revise throughout in manuscript
  11. Histological figures look not good. I suggest to replaced the figure having scale bar and good resolution.
  12. In figure 6 I suggest to include standard and then compared the results with that of standard compounds. Similarly, for HPLC how author compared the data. Need a standard compounds and detailed about data analysis in figure analysis.

Author Response

Reviewer 3

Open Review

Comments and Suggestions for Authors

The manuscript entitles “Antioxidant and anti-inflammatory effects of Peganum harmala extracts: An in vitro and in vivo study” investigated that the potential of Peganum harmala extract, their bio constitutes, and finally their bioactivities both in-vitro and in-vivo.  The subject frame of the work is well constructed. So, in this respect and this article should be contributed to present research. I recommended this work re-consideration for publication after the following major revisions.

  1. The manuscript written very poorly, sometime its difficult to fellow the story. I recommend to use of proper English grammar. Also, there are several typographical mistakes as well in whole manuscript. Therefore, the author’s thoroughly careful check the language and typo mistake to minimize the error.

            Response: Manuscript has been checked and typos corrected

  1. The abstract should be beginning with a sentence about the background of concept and the aims as well as novelty of study should be mentions. What exactly is the novelty of this study? The abstract is poorly written and should be improved. Abbreviations must be avoided in abstract. Parenthesis should be avoided in abstract - this is poor writing. Please improve.

Response: Abstract has been revised

  1. Introduction; Check and format the citations in the whole manuscript. Also, Appropriate references must be provided to explained the background, what is already done and why this study carried out. Other vise the novelty of this research is still poorly presented. This is important especially for the high IF journals. The scientific style should be used. What exactly is the aim of this work? Hypothesis statement is missing in the introduction section.

Response: Introduction has been revised and new references added.

  1. Introduction; Introduction section should be elaborated with recent articles on antioxidant and anti-inflammatory activities. Also about extract preparation as well as the importance of medicinal plant is missing. For example; the recommended literature are as fellow; https://www.sciencedirect.com/science/article/pii/S0045206815300031;
    https://www.sciencedirect.com/science/article/pii/S0141813021011818;
    https://www.sciencedirect.com/science/article/pii/S0278691517301035;
    https://www.sciencedirect.com/science/article/pii/S2213343721002670;
    https://www.sciencedirect.com/science/article/pii/S1570023218305749;

Response: Recent articles have been added to the revised manuscript as suggested by the reviewer

  1. Material and methods; The whole M&M section is poorly written and must be substantially rewritten and improved. The methods are not properly referenced and are not possible to follow, reproduce and verify. This is serious drawback of this research work. You must be precise when writing protocols (M&M section); everyone should be able to repeat your experiments, after this paper is published, and gain the same results. Serious lack of relevant information or poorly written Materials and methods section are always an alarm form me, as a reviewer. For example, Materials are poorly described. Furity and product codes of all these materials should be provided. Materials are poorly described. Product codes should be provided. Appropriate references are missing. Who exactly is the Author of the methods applied? Are these methods valid/correct? Without sufficient information it is hard to reproduce the results, follow, and compare the idea and the obtained results with the literature. Appropriate references to the methods should be provided.

Response: The whole M & M section has been revised with proper references and details

  1. The statistical findings have to be given in the text such as (p<0.05) or (p>0.05).

Response: Corrected

  1. Results and discussion; General remark to the discussion - In my opinion, the discussion provided by Authors is difficult to follow and verify due missing critical details in the methodology section. Due to poorly described material and poorly presented methods, I am not able to follow and properly review the discussion.

Response: Discussion section has been revised.

  1. All figures are of poor technical quality and not suitable for publication, especially in a high reputed journal. Font size and kind is too small and must be unified in all figures. Small writings are unreadable. All figures must be self-explanatory. Axis titles are poorly presented or absent. Units are missing. Are the data presented in figures significantly different? At least error bars should be shown.

Response: Figures have been replaced with better quality images

  1. Details about the statistical testing are needed.

Response: Phytochemical and antioxidant activities data are expressed as mean ± SD. Statistical differences between the control and treatments were analyzed using one way and two-way

analysis of variance (ANOVA) for in vitro and in vivo experiments respectively, followed by Dunnett’s Multiple Comparison test. Graphs were plotted using graph pad prism.

  1. I suggest first time write full name rather than abbreviation; revise throughout in manuscript

Response: Corrected as suggested

  1. Histological figures look not good. I suggest to replaced the figure having scale bar and good resolution.

Response: Replaced with better quality images

  1. In figure 6 I suggest to include standard and then compared the results with that of standard compounds. Similarly, for HPLC how author compared the data. Need a standard compounds and detailed about data analysis in figure analysis.

Response: In Fig 6 mass spectras were compared with published literature and considered as tentative identification. Furthermore, we conformed these compounds on HPLC using external standards by comparing the retention times. We also added an overlay HPLC chromatogram between Standards and Plant fraction as Fig 9 for better understanding.

Round 2

Reviewer 2 Report

In accordance with the reviewers' suggestions, the changes made by the authors   bring clarifications and improve the quality of the manuscript.

Reviewer 3 Report

no further comment